# Accelerating Policy Gradient by Estimating Value Function from Prior Computation in Deep Reinforcement Learning

**Md Masudur Rahman & Yexiang Xue**
Department of Computer Science
Purdue University
West Lafayette, IN 47907, USA
`{rahman64,yexiang}@purdue.edu`

## Abstract

This paper investigates the use of prior computation to estimate the value function to improve sample efficiency in on-policy policy gradient methods in reinforcement learning. Our approach is to estimate the value function from prior computations, such as from the Q-network learned in DQN or the value function trained for different but related environments. In particular, we learn a new value function for the target task while combining it with a value estimate from the prior computation. Finally, the resulting value function is used as a baseline in the policy gradient method. This use of a baseline has the theoretical property of reducing variance in gradient computation and thus improving sample efficiency. The experiments show the successful use of prior value estimates in various settings and improved sample efficiency in several tasks.

## 1 Introduction

Reusing past computations has brought tremendous success in many machine learning fields, including computer vision (e.g., pre-trained models on ResNet) and natural language processing (e.g., large language models like GPT-3). These techniques allow for the creation of practical tools in real-world problems where task-specific data is scarce. Reinforcement learning (RL) provides algorithmic advantages and works on dynamically changing datasets in many cases in contrast to the static datasets used in supervised learning. However, this comes at the cost of requiring many samples. Despite this, RL has shown tremendous breakthroughs in recent years, particularly when large amounts of data (e.g., in simulations or games) are available (Silver et al., 2016; Vinyals et al., 2019). However, using RL in many real-world tasks is challenging, partly due to the scarcity of data (i.e., environment interaction) that most RL algorithms require.

Off-policy algorithms provide a mechanism for reusing data. However, their performance can be unstable, and they often take longer runtimes due to their sequential nature of environment interaction (e.g., DQN). In contrast, on-policy policy gradient methods provide a simplified RL objective, allowing for efficient use of computing resources and faster runtimes and thus preferred choice in many real-world settings, including robot learning (Peters & Schaal, 2006). However, this comes at the cost of being less sample efficient in many cases.

This paper investigates the use of prior computation to estimate value function to improve sample efficiency in on-policy policy gradient methods. There has been interest in reusing past computations in RL settings in many forms, including prior collected data (Agarwal et al., 2022), expert demonstration (Ho & Ermon, 2016; Rajeswaran et al., 2017), offline data and fine-tuning (Julian et al., 2020; Singh et al., 2020; Kostrikov et al., 2022; Lee et al., 2022c;b). In contrast, our approach is to estimate the value function from prior computations, such as from the Q-network (Q function) learned in DQN or the value function trained for different but related environments. Finally, we leverage the estimated value and integrate it with the policy gradient method (e.g., PPO). As training progresses, the emphasis on the prior estimated value function weans out, and finally, the new

algorithm becomes free from any such prior computation. In this way, the RL training can get a jump-start and thus potentially achieve improved sample efficiency.

Specifically, we provide a mechanism to estimate a value function from the Q function that is learned during the training of DQN. By definition, the Q function and value function depend on a policy; thus, the pre-trained Q function cannot be directly used for policy gradient. Instead, we use the value estimate as a baseline which can reduce variance in gradient computation and thus improve sample efficiency. Our estimate of the gradient in the policy gradient method with a value baseline is theoretically unbiased as the baseline does not depend on the action. In one setting, we leverage this property and combine the value estimate from the pre-trained Q function and the value estimate of the current on-policy. We still want to include the current value estimate, as it might be necessary in the case of generalization, where the current environment varies from the pre-trained environment. For example, the gravity of the Lunar Lander might be different on Earth and Mars. Therefore, we use the pre-trained Q function on Earth and can deploy the policy gradient method on robots on Mars. This approach still provides an unbiased gradient estimation estimate, with lower variance and thus better sample efficiency theoretically. The pre-trained value function or Q-function can come from any policy of any algorithm or can be given manually when feasible.

We conducted experiments to demonstrate the successful use of prior value estimates in various settings. We leveraged the value function for the policy gradient method Proximal Policy Optimization (PPO) (Schulman et al., 2017) for discrete and Robust Policy Optimization (RPO) (Rahman & Xue, 2022) for continuous action. Specifically, we repurposed the Q-function from off-policy DQN in the same environment, repurposed the Q-function from off-policy DQN in different environments, repurposed the value function from different environments, and repurposed the value function from the same environments. Our results demonstrate the usefulness of using past computations while improving sample efficiency in several tasks. Furthermore, our empirical results support the use of pre-trained value functions in policy gradient, which has a theoretically sound explanation for such usage.

The results of the experiments show that our RRL-PPO algorithm, which used past computation, performed better than the TBR-PPO (Tabula Rasa, trained from scratch) baseline in several settings. Specifically, our method performed better in the LunarLander, BeamRider, and Breakout Atari environments, where a pre-trained Q-network from a DQN trained on the same environment is used for the RRL method. The RRL-PPO quickly solved LunarLander, and performance improvement in BeamRider and Breakout was observed. In the windy LunarLander environment, RRL-PPO performed better than TBR-PPO, which trained from scratch, where the pre-trained Q-network was in a different, non-windy LunarLander environment.

In a different reward function setting, using a value function trained in one task (walker stand) improved performance in a related task (walker run) with a small but consistent gain in performance. When using a previously estimated value function in a subsequent run where the source and target tasks and underlying algorithm are the same, we observe that our RRL-RPO agent reaches a high score in just 2 million timesteps of training.

In comparison, the base RPO takes 8 million timesteps to reach the same score. Overall, the results demonstrate that incorporating past Q-function computed by a DQN improves the performance of policy gradient methods and can improve the sample efficiency of on-policy methods. Our suggestion for reincarnating reinforcement learning for the policy gradient method is to pre-trained general value functions for domains with multiple downstream tasks.

## 2  PRELIMINARIES AND PROBLEM SETTINGS

**Reinforcement learning (RL)** is a learning framework where an agent interacts with an environment to maximize a cumulative reward signal. A policy maps states of the environment to actions that determine the agent's behavior. The goal is to find an optimal policy that maximizes the expected cumulative reward. The underlying assumption is that the reward signal is enough to guide an expected behavior.

In RL, the task often follows a **Markov Decision Process (MDP)**, which consists of a tuple $\mathcal{M} = (\mathcal{S}, \mathcal{A}, \mathcal{P}, r)$. At each timestep $t$, the agent in a state $s_t \in \mathcal{S}$ takes action $a_t$ from the set of actions

$\mathcal{A}$. The agent then receives a reward $r_t$ and the environment transitions to a new state $s_{t+1} \in \mathcal{S}$ according to the transition probability $P(s_{t+1}|s_t, a_t)$.

Model-free algorithms do not model the environment's reward and dynamics. Instead, they try to learn the policy directly from the collected trajectory or experience. **Off-policy** algorithms (e.g., DQN) leverage past computation by storing the experience in a replay buffer. A Q-function is learned from the experience and then a policy is extracted when the agent needs to perform. Due to the reuse of past experience, these methods usually achieve better sample efficiency. However, the runtime for these methods is often high and algorithmic constraints can prevent the use of parallel processing for faster training.

In contrast, an **on-policy** method, such as the policy gradient method, directly learns a policy and optimizes the reward objective instead of learning a Q-function. Due to the on-policy nature, past data is often discarded, and the agent's policy is updated on newly collected data. This process can lead to sample inefficiency, which might prevent it from being used in many tasks. This paper aims to improve the sample efficiency of such policy gradient methods through prior computation in RL training.

**Value functions** and **Q-functions** are fundamental building blocks of many RL algorithms. They are a key component of RL, as the value function estimates the expected cumulative reward for a given policy and state. On the other hand, the Q-function, also known as the action-value function, estimates the expected cumulative reward for a given policy, state, and action. By definition, value functions and Q-functions are related; however, their use in algorithms may vary and result in various forms of algorithms.

**Problem Settings**: We consider repurposing prior computation RL similar to the concept of reincarnating reinforcement learning (RRL) (Agarwal et al., 2022). In our setting, an agent is trained on a source task, and then this computation is transferred to a target task. The computation can take many forms, and in this paper, we focus on reusing a previously computed value function to improve sample efficiency in the target task. Specifically, we propose a method to incorporate a previously estimated Q-function or value function as a baseline in a policy gradient method to reduce the variance of the gradient computation. We experiment with four different settings in which we leverage the estimated value function and use it as a baseline.

## 3   METHOD: REINCARNATING VALUE ESTIMATION

In this section, we discuss the theoretical motivation of our approach and demonstrate how we leverage the value estimate from the prior computation. In the policy gradient method, the goal is to learn the gradient of the policy that has the reward in its objective, which is maximizing cumulative future reward. Thus, if we can compute the gradient, we can optimize the policy using samples collected from the current policy. The gradient estimation $\tilde{g}$ from samples in the policy gradient method (Sutton & Barto, 2018) is as in equation 1.

$$\tilde{g} = \frac{1}{m} \sum_{i=0}^{m} \nabla_\theta \log P(\tau^{(i)}; \theta) R(\tau^{(i)}) \tag{1}$$

where, $\nabla_\theta \log P(\tau^{(i)}; \theta)$ can be further decompose into $\nabla_\theta \log P(\tau^{(i)}; \theta) = \sum_{t=0}^{H} \nabla_\theta \log \pi_\theta(a_t^{(i)}|s_t^{(i)})$, where $H$ is the horizon of an episode. The $m$ is the number of trajectory that used to estimate the gradient. The $R(\tau^{(i)})$ is the cumulative reward of a trajectory $\tau^{(i)}$.

This gradient estimation method in equation (1) provides a practical way to estimate the gradient from the sample trajectory collected by the current policy. However, although this provides an unbiased estimate, the gradient estimation can have high variance (Greensmith et al., 2004). A simple and effective way to reduce the variance is to use a baseline, $b$, as in equation (2).

$$\tilde{g} = \frac{1}{m} \sum_{i=0}^{m} \nabla_\theta \log P(\tau^{(i)}; \theta)(R(\tau^{(i)}) - b) \tag{2}$$

Several methods have been proposed to be used as a baseline (Greensmith et al., 2004) to reduce the variacne of gradient estimation. It can be shown that equation (2) results in still an unbiased estimate as long as the baseline does not depend on the action. This property enables various quantities to be used as a baseline, which can be unbiased (theoretically sound). An effective choice of the policy can be a state-dependent baseline, such as a value function $b(s_t) = V(s_t)$. For a policy $\pi$ the function can be defined as

$$b(s_t) = V^\pi(s_t) = \mathbb{E}[r_t + r_{t+1} + ... + r_{H-1}]$$

The value function can be estimated using the Monte Carlo Estimate of sampled rewards. In deep RL, a neural network is used to approximate the value function by regressing with the Monte Carlo Estimation. This Monte Carlo estimate is unbiased, however, it can take a lot of samples for a better estimate due to the nature of the RL setup, as it requires to have the same state on multiple trajectories to have a better estimate of the value on that state, which is challenging, especially in high-dimensional environments.

Due to the poor Monte Carlo estimate, the overall value function can be near to random, which might badly impact the policy learning, as bad exploration can lead to a bad overall policy and the policy can show a myopic bias, which eventually can be detrimental to the overall policy learning.

In our paper, we propose leveraging proper computation to estimate the value function which can help to learn a better value function using the above method. We combine the prior value function with the new to generate the final policy value function as in Equation (3).

$$b(s_t) = (1 - w_t) * V_\pi(s_t) + w_t * V_\pi^{prior}(s_t) \tag{3}$$

Here $V_\pi(s_t)$ is the value estimate of the current policy and $V_\pi^{prior}(s_t)$ is the value estimated from past computation. The weaning off parameter $w_t$ is the weight between 0 to 1 ($w_t \in [0, 1]$), which determines the amount of focus on the pre-computed value estimate. Note that, the $V_\pi^{prior}(s_t)$ is computed from a fixed quantity that is expected to be given and computed from "some" past computation. The hyperparameter $w_t$ depends on the total timestep of training and usually decreases in value as training progresses. The intuition is that this process will make up for the initial bad estimate of the value estimate of the current policy and give it a jumpstart. Note that, the baseline $b(s_t)$ in equation (3) still does not depend on the current action. Thus this choice of baseline would result in an unbiased estimation of the gradient. However, this value estimate as a baseline can potentially reduce the variance in the gradient estimate.

The current value function does not depend on action that generate the trajectory. The value estimate from prior computation is fixed in the target task and does not depend on the action of the current policy either. Thus, their linear combination in the baseline $b(s_t)$ does not depend on the current action. Therefore, the gradient is unbiased.

Thus our method provides a theoretically sound way to leverage prior RL computation in the policy gradient-based method.

**Values Estimate for Prior Computation**. We now discuss various ways that prior computation can be used to estimate the value function $V^{prior}$.

*Q-function to Value Estimate*. The use of Q-function is essential to many off-policy model-free algorithms to learn policies such as DQN. In function approximation, a Q-function is represented as a neural network and is learned from the agent's experience. In this setup, we assume that a Q-function is given which gives a Q-value given a state and action.

The value function and Q-function is related. For discrete action the value estimate can be computed as in equation 4.

$$V_\pi^{prior}(s_t) = \sum_{a_i \in A}^{|A|} [\pi(a_i|s_t)Q(s_t, a_i)] \tag{4}$$

where, $A$ is the action set. For finite discrete action, the value estimate $V_\pi^{prior}(s_t)$ can be computed using equation (4). Here, $\pi$ is the current policy which generates the action probability. For the continuous action case, the value estimate is defined as integrating the Q-value w.r.t the action as in equation. On the other hand, estimating the value from Q-function can be challenging for the continuous case. In continuous action, the number of possible actions can be infinite; thus, the quantifying equation for the value function can be intractable. However, in this paper, we demonstrate our method for discrete action space only.

*From value estimate to value function.* This process assumes the use of a similar value estimation for the value function in the target task.

$$V_\pi^{prior}(s_t) = V_{\pi_{past}}(s_t) \qquad (5)$$

Here, for different environment dynamics, $V_{\pi_{past}}(s_t)$ is the value function of a previous policy gradient training in an environment where the environment is different for the current one by different dynamics only. On the other hand, in case of different reward functions (stand, run), the $V_{\pi_{past}}(s_t)$ is the value estimate that learned in an environment where the reward function is different but other MDP properties such as dynamics are the same. In this case, the past value estimate is from similar algorithms where it uses a value function estimate directly.

This paper evaluates the value estimation from prior computations in four different settings. In Setting 1, a pre-trained Q-network that was trained using DQN is given where the source and target environments are the same. Setting 2 uses a pre-trained Q-network from a different source environment than the target. In Setting 3, a PPO agent is trained on the walker stand task, and the trained value function is given to the RRL agent; the evaluation is done in a different environment, the walker run. Finally, setting 4 demonstrates how a value function can be reused for the same algorithm on the same task. Overall, this paper primarily focuses on improving sample efficiency in the policy gradient method through prior computation.

## 4 EXPERIMENTS

We aim to show how using value estimation from previous calculations can speed up policy learning in policy gradient-based, on-policy methods. Policy gradient methods are flexible and often have fast runtime, but their sample efficiency is a concern for many tasks. We demonstrate that previous calculations, even from different algorithms (e.g., DQN) and environments, can be used to improve sample efficiency.

**Baselines**. We name our reincarnated version of the agent with the RRL (Reincarnating Reinforcement Learning) prefix. For the on-policy algorithm, we use PPO (Schulman et al., 2017), labeled as **RRL-PPO**, for discrete actions and RPO (Rahman & Xue, 2022)[1], labeled as **RRL-RPO**, for continuous actions. The PPO algorithm trained from scratch in the target task is named **TBR-PPO** (Tabula Rasa). The names RRL and TBR are inspired by (Agarwal et al., 2022).

### 4.1 SETUP

**Environments** We conducted experiments on continuous control tasks from four reinforcement learning benchmarks: OpenAI Gym (Brockman et al., 2016) Lunar Lander in various generalization settings, Atari (Bellemare et al., 2013) BeamRider and Breakout and DeepMind Control Suite (Tunyasuvunakool et al., 2020) Walker stand and run. Figure 1 illustrates the environments used in our experiments. We evaluated a diverse range of environments, including vector and image observations with discrete and continuous action spaces. For the DQN training, we used the "NoFrameSkip" version of the Atari environments. Regarding implementation details, we trained DQN on the Gym implementation of BeamRiderNoFrameskip-v4 and BreakoutNoFrameskip-v4. However, Envpool (Weng et al., 2022) offers a way to access Atari environments in parallel. Policy gradient methods like PPO can take advantage of this parallel environment access for more efficient rollouts and faster training. DQN-based methods, on the other hand, are not typically optimized for this parallel advantage. In the Q-function to Value function estimation experiment for Atari, we use a pre-trained Q-function where the environments were implemented in different library architectures. We then leverage it with the policy gradient method and take advantage of the parallel power of the library. This process allows training PPO in 1.5 hours for 10M timesteps instead of 8 hours for 10M timesteps in our experimental setting. This cross-library implementation is essential as the library of environment models is evolving rapidly.

**Implementation** Implementing RL algorithms and managing hyperparameters are crucial for comparing different algorithms. In this paper, we use the CleanRL library (Huang et al., 2022a; 2021) to implement all of our agents, including our baselines. Unless otherwise specified, we use the same

---

[1]Due to better empirical performance, CleanRL recommends RPO over PPO for continuous actions (Huang et al., 2022b)

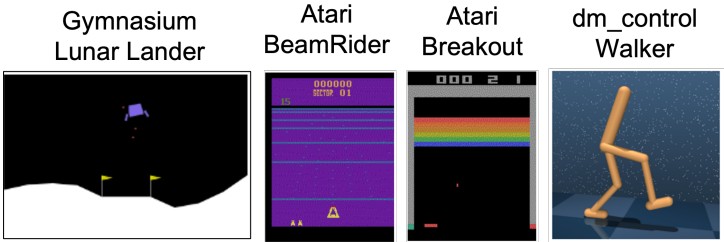

Figure 1: Snapshots of the environments used in our experiments. The Lunar Lander environment has options to change the dynamics, such as enabling wind and changing gravity, which we leverage for generalization experiments. It has vector-based observations and discrete actions. The Atari BeamRider and Breakout consist of image-based observations and discrete actions. The Walker domain is from the DeepMind Control Suite, where we use the stand and run tasks. It has vector-based observations with continuous action spaces.

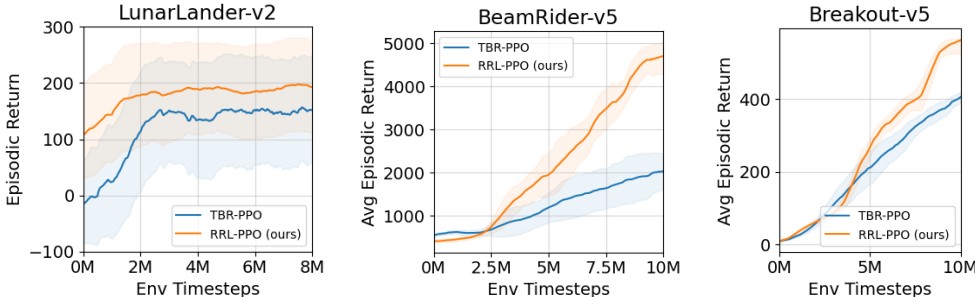

Figure 2: Result comparison in Setting 1. The performance of our RRL-PPO algorithm in LunarLander, BeamRider, and Breakout Atari environments is better compared to the TBR-PPO baseline. The RRL-PPO quickly solved LunarLander, and the performance improvement in BeamRider and Breakout increased after 2.5M and 5M timesteps, respectively. The DQN algorithm also performed well but had slower runtime. Our results demonstrate that incorporating past Q-function computed by a DQN improves the performance of policy gradient methods.

hyperparameters that are set as default in the CleanRL library. We use DQN as the off-policy algorithm to learn the pre-trained Q-function. To account for stochasticity, we run experiments for several seeds and report the mean and standard deviation in the learning curves.

The random seed runs are 10 for LunarLander and Walker run and stand, 5 for Atari Breakout, and 4 for BeamRider (fewer runs due to their smaller standard deviation across runs).

**Evaluation Settings** We evaluate the value estimation from prior computations in several settings. Note that when a pre-trained agent is provided, we only use a single seed run (the best performing) for value estimation to ensure a higher quality prior computation of the Q-function and value function.

*Setting 1.* A pre-trained Q-network is provided that is trained using DQN in a discrete action space. The underlying environments are the same for the pre-trained (source) and target environments for the policy gradient that uses the estimated value. This setting enables prior testing computation with different algorithms. In this setting, we experiment with LunarLander (no wind), Atari BeamRider, and Breakout. As the environment is the same for both the source (pre-trained) and target, we did not use a weaning rate schedule and instead kept a fixed $w$ for the entire training of the RRL version of the agent. We set the fixed value to $w = 0.9$ in these experiments.

*Setting 2.* In this setting, a pre-trained Q-network is provided from a different source environment than the target. Specifically, a DQN agent is trained on LunarLander without wind, and its Q-network is given to the RRL for value estimation. The target task is LunarLander with wind enabled, which becomes challenging to learn from scratch due to the changes in dynamics where wind creates

additional challenges for the lunar landing. In this setting, the agent is evaluated on a different task and algorithm.

We use a starting learning rate of $w = 0.5$ and reduce it by $0.1$ after every $1M$ timestep. Thus, the value is completely weaned off after around $5M$ timesteps; then, the RRL method only uses the newly learned value function.

*Setting 3.* In this setting, the task is different but on the same domain. The walker stand environment is used as the source, and the walker run environment is used as the target. A PPO agent is trained on the walker stand, and the trained value function is given to the RRL agent. The RRL agent leverages the learned value, and the evaluation is done in a different environment, the walker run. This setting evaluates how pre-computation from the same algorithm can be leveraged to solve a new task on the same domain. Here we assume that the environment domain, dynamics, action space, and state space remain the same; however, the reward function is different, that is, stand to run. An implicit assumption is that the source task should be related to the target task. For example, the stand skill is essential for the agent to learn the run. We set the starting learning rate of $w = 0.5$ and reduce it by $0.1$ after every $1M$ timestep. The value is completely weaned off after around $5M$ timesteps; then, the RRL method only uses the newly learned value function.

*Setting 4.* In this setting, we demonstrate how a value function can be reused for the same algorithm on the same task. This scenario can occur when we train an agent for a few random seed runs but later decide to rerun it. For example, there may be an issue with logging data or a power shutdown that caused the past run to fail. This scenario might be useful when deploying RL to real-world tasks where such interruptions are commonplace. Additionally, this setting also demonstrates the need for a better value function estimation at the start of agent training on a task. The starting learning rate is set to $w = 0.5$, and the weaning interval is $1M$.

## 4.2 RESULTS

We now discuss the results of our experiments in different settings.

*Results of Setting 1.*

Figure 2 shows the result of this setting. We observe that our RRL-PPO performs significantly better in all three environments than the TBR-PPO. In LunarLander, the performance quickly reaches 200, which is considered the task to be solved. Here, the DQN run used for RRL also achieved the solved state for LunarLander. On the other hand, the TBR-PPO could not reach the solving mark on mean performance by 8M timestep.

Similarly, for the two Atari environments, our RRL-PPO performed better compared to the TBR-PPO baseline. In the BeamRider environment, RRL performed comparably up to 2.5M timesteps with TBR. However, after that, the performance improved drastically for RRL, reaching about 4500 compared to just above 2000 for TBR. The policy network was also implemented using a CNN-based neural network. Thus it needs some training time to output meaningful actions, which might explain the initial comparable performance.

Similarly, for Breakout, the performance started to become better for RRL after about 5M timesteps. The variation in the start time to show a difference in performance varies across

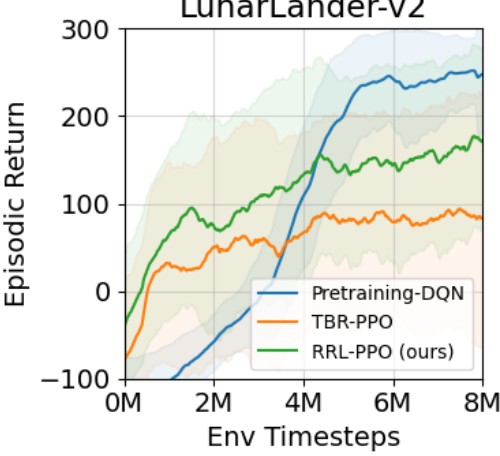

Figure 3: Result comparison in Setting 2. The performance of RRL-PPO in a windy LunarLander environment is better than TBR-PPO trained from scratch. The DQN performance in a non-windy environment and its Q-network is used in RRL-PPO. The results demonstrate that RRL-PPO performed better than DQN, even in a more challenging environment, during the first 4M timesteps.

environment tasks. This result is expected as learning policy might vary based on varying complexity across tasks. Additionally, the performance improvement of BeamRider is higher compared to the difference in Breakout for RRL-PPO. This result suggests that the gain also depends on the quality of the pre-trained Q-function. A better pre-training, in general, should result in a better performance gain. However, a bad-quality pre-trained Q-function has the potential to worsen the performance. In our setting, a weaning factor can be used to prevent such worsening. Thus, we recommend tuning the weaning factor $w$ based on task complexity and the quality of the pre-trained Q-network.

In these environments, the DQN performed better than both agents when trained for 10M timesteps. However, the runtime for the DQN was much slower (almost 10 times slower than PPO's run). We see that the performance of the policy gradient method PPO can be significantly improved (e.g., in BeamRider) using past Q-function that is computed using a DQN agent in the past.

*Results of Setting 2.* Figure 3 shows the results in this setting where the source and target environments are different, and also, a pre-trained Q-network is used. We also provide the Pre-trained DQN curve.

The results show that our method RRL-PPO performs better than the TBR-PPO, which is trained from scratch in the windy LunarLander environment. Due to wind, the overall performance of on-policy methods TBR and RRL is lower compared to the non-windy LunarLander. Still, our method performs better than training from scratch by a large margin in mean performance value. Here, the DQN performance is on the source environment where the wind was disabled; thus, they are not directly comparable as they were in an easy environment. However, we still show it to demonstrate the performance difference. Even with

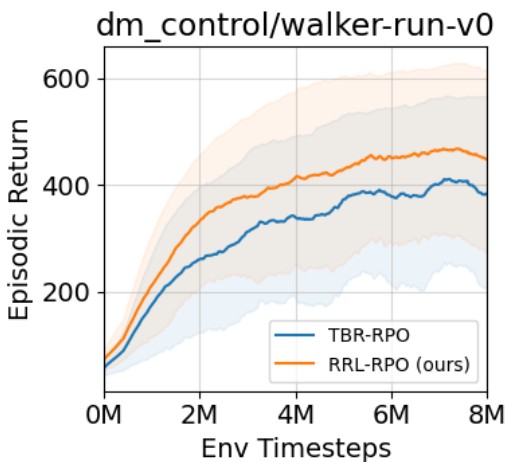

Figure 4: Result comparison in Setting 3. The result shows that using a value function trained in one task (walker stand) improves performance in a related task (walker run). The gain in performance is small but consistent across timesteps and persists even after the past value estimate is phased out at around 5M timesteps.

this setup, the DQN performance was bad at the first 4M timesteps, whereas our RRL-PPO method performed better than DQN in a more challenging environment.

*Result of Setting 3.* Figure 4 shows the results in this setting. We observe that using a past-computed value function helps improve the performance in a different task on the same domain.

Here, the value function trained in the walker stand task helps the performance in the walker run. However, the performance depends on how the reward function for two tasks is related. Thus, we observe a relatively small gain in performance. A more related task might result in a better performance gain. Nevertheless, the performance gain is consistent across timesteps and remains after the past value estimate wean off at around 5M timesteps.

*Results of Setting 4.* Figure 5 shows the results in this setting. When the source and target domain are the same, the value function estimated from RPO's previous run can help improve the performance in a subsequent run at a later time.

The walker stand performance improved compared to the RPO agent, which was trained from scratch. This result shows the potential of achieving a better value function that can improve the sample efficiency of the on-policy method. Our RRL-RPO agent reached a score above 800 in just 2 million timesteps of training, while the base RPO could not reach 800 even after 8 million timesteps. Thus, a well-pre-trained value function can help achieve the task with better sample efficiency for any real-world task.

## 5  RELATED WORK

Behavior Transfer is proposed for transfer learning in reinforcement learning by leveraging pre-trained policies for exploration (Campos et al., 2021). Simulation computation has been leveraged to improve real-world tasks by introducing domain randomization to create diverse training data (Akkaya et al., 2019). These approaches consist of creating diverse data in one domain, such as simulation, and performing in different domains, such as real-world tasks. External supervision has been leveraged to scale up by using prior demonstrations and heterogeneous data collected from prior collected data (Lu et al., 2022).

Fine-tuning provides ways to incorporate pre-learning to transfer to target tasks, such as training a policy using offline data and then finetuning it for the target task (Julian et al., 2020; Singh et al., 2020; Kostrikov et al., 2022; Lee et al., 2022c;b). Pre-trained representations are also explored for accelerating sample efficiency, including unsupervised pre-training without the reward label and with access to an exploration scheme or replay buffer (Touati & Ollivier, 2021), and adapting to new tasks. Unsupervised pre-training has been leveraged to learn behavior without a reward function (Liu & Abbeel, 2021; Touati & Ollivier, 2021). However, these methods often require data from the same underlying task with the same dynamics. In contrast, we have demonstrated the use of past computation in several settings, including various dynamics and reward functions.

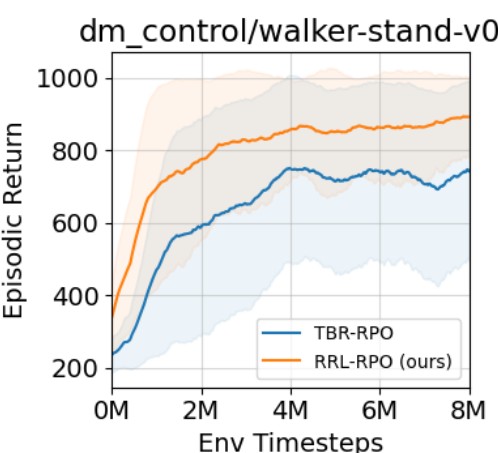

Figure 5: Result comparison in Setting 4. The result demonstrates that using a previously estimated value function can improve performance in a subsequent run when the source and target tasks are the same. The walker stand performance is better than when using the base RPO agent trained from scratch, showing the potential for improving sample efficiency of on-policy methods. The RRL-RPO agent reaches a high score in just 2 million timesteps of training, while the base RPO takes 8 million timesteps to reach the same score.

A general value function can be trained by leveraging the process of multidomain scalings, such as learning from internet-scale video (Baker et al., 2022). The value function can then be added based on our method proposed in this paper to leverage for the target task. Our method provides a simple way to incorporate such value function into policy gradient methods through a baseline. Kick-starting a previously trained policy has been leveraged to improve sample efficiency in multi-task benchmarks (Schmitt et al., 2018). Collected data from prior computation has been leveraged (Agarwal et al., 2022; Lee et al., 2022a). Such methods provide a simple way to integrate past computations. However, the quality of the collected data can severely impact the performance gain. Furthermore, the collected data can also prevent the use of the collected data in many settings, especially in on-policy setups where it requires collecting data using the latest policy to be theoretically sound.

In this paper, we propose improving the performance of on-policy algorithms where such previously collected data cannot be integrated directly. Another issue with this prior data collection is using it in new settings. Due to slight environmental changes, the collected data might be confounded and prevent learning a suitable policy at the kickstart and eventually fail to learn a good policy. This setup is commonplace in real-world applications. However, a better-learned Q-function or value function can be free from any dynamics changes or confounders and efficiently transported to the target task. On the other hand, the storage requirement of collected data is high, especially in high-dimensional environments where a large amount of data is required to achieve a meaningful kick-start on the target task. QDagger (Agarwal et al., 2022) has access to both the suboptimal source policy and some data from the prior computation. In contrast, we need to access the Q-function for DQN or value function without data from the prior computation. Furthermore, the use of the replay buffer

limits the ability to be used with policy gradient-based algorithms as it cannot use the replay buffer data in a theoretically sound way.

To reincarnate reinforcement learning, we recommend building a general value function similar to large pre-trained models (e.g., BERT, GPT-3) for a benchmark. For example, a general pre-trained value function can be for the DeepMind Control benchmark for each domain. Once the state and action space is unified for a robotic domain, learning a general value function can provide an advantage in solving the underlying tasks. New tasks can be added; for example, the value model can be trained on the stand, walk and run, and then later asked to solve trot (for dog robot in $dm_control$ (Tunyasuvunakool et al., 2020)). This process will allow the designing efficient algorithms for end tasks that can leverage the general value function. With the capability of repurposing computation to solve new tasks, the reincarnation process will lead to solving more practical problems using reinforcement learning and enable deployment in the real world.

Demonstration data has been leveraged to accelerate learning in new tasks in various formats such as from specific tasks (Ho & Ermon, 2016; Rajeswaran et al., 2017) and other related tasks (Singh et al., 2020). Another form of transferring knowledge is to reuse previously trained models (Sun et al., 2022) as representations. An abstract representation is trained to accommodate changing features in the observation space (Sun et al., 2022). World models have been trained to accelerate sample efficiency in multi-source transfer settings (Sasso et al., 2021).

In contrast, our method leverages past computation for estimating a value function. This component is then used as a baseline for the policy gradient method, potentially reducing the variance in gradient computation. We have demonstrated the use of such computation in four settings. However, many approaches discussed above can potentially be leveraged to estimate the value function, such as from an offline dataset, replay buffer data, or past demonstrations. As long as we can estimate the value function, we can leverage it using our method to solve the target task.

## 6  CONCLUSION

In this paper, we proposed a method to improve sample efficiency in on-policy policy gradient methods by utilizing prior computation to estimate the value function. By combining a value estimate from prior computations with a new value function for the target task, we were able to use the resulting value function as a baseline in the policy gradient method. This approach could reduce variance in gradient computation and improve sample efficiency. Our experiments showed that this method is effective in various settings and can significantly improve sample efficiency in several tasks. In particular, we demonstrate the effectiveness of using prior computation in four settings. Overall, this paper highlights the potential of leveraging prior computation to improve sample efficiency in on-policy RL and may pave the way for more efficient and effective policy gradient algorithms in various practical settings. Our recommendation for reincarnating reinforcement learning is to build a general value function similar to large pre-trained models (e.g., BERT, GPT-3, ResNet) for domains with various downstream RL tasks.

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
