# OpenReview forum: "Accelerating Policy Gradient by Estimating Value Function from Prior Computation in Deep Reinforcement Learning"
_ICLR.cc/2023/Workshop/RRL — RRL 2023 Poster_

### Official Review · Reviewer_pDd2 · 2023-02-27
**A valid way to increase sample efficiency but requires further novelty and validation**

**Rating:** 2
**Confidence:** 3

**Review:**

This paper improves sample efficiency from the pre-training perspective by incorporating prior computations as the baseline values in PPO. The topic is interesting and on the trend. Based on the experiment, it seems that the method did fasten the training processes.

Pros:
1. The experiments settings are thorough.
2. The paper is written to make readers easy to understand
3. The idea of including pre-trained computations into baselines is interesting.

Cons:
1. The idea of improving sample efficiency of gradient methods from the perspective of pre-training is interesting. However, in PPO with baseline, the idea of using estimated value function as the baselines has been proposed before. The paper simply proposes a weighted combination of current value function estimation and the one from prior computations.
2. In the method, the paper presents REINFORCE, while in the experiment they adopt PPO. The inconsistency causes confusions. Furthermore, in line 188-189 the authors mentioned the *Q-function to Value Estimate* is demonstrated for discrete action spaces, while in the experiment the action space of all the environments are continuous. The authors left out the details for the adaptation.
3. Though the experiment settings are thorough, this paper only compares their method with one baseline method.

In summary, in combination with the experiments I agree that the paper proposes a valid idea to improve sample efficiency, and the perspective is interesting. However, the paper has some spaces for improvement in clarifying confusions, and the idea is yet to be confirmed on more SOTA methods. Therefore I propose a marginally acceptance.

---

### Official Review · Reviewer_7CFD · 2023-03-02
**Summaries of reviews**

**Rating:** 1
**Confidence:** 4

**Review:**

The paper proposes to re-use the value function learnt in a different environment as the baseline in PPO. The paper claims doing so reduces variance of the estimates and therefore improve performance.

There are two main issues with the paper:

1. Lack of support for their main claim, which is the lower variance of the estimates. Can the authors actually compute the variance and show that it is lower?

2. Lack of more comprehensive literature review. Re-using networks pre-trained in related environment is an active research area in offline RL. For example, please see https://proceedings.neurips.cc/paper/2020/hash/4496bf24afe7fab6f046bf4923da8de6-Abstract.html